# Association between COVID-19 Vaccine Side Effects and Body Mass Index in Spain

**DOI:** 10.3390/vaccines9111321

**Published:** 2021-11-15

**Authors:** Isabel Iguacel, Aurelio Luna Maldonado, Aurelio Luna Ruiz-Cabello, Marta Casaus, Luis Alberto Moreno, Begoña Martínez-Jarreta

**Affiliations:** 1Faculty of Health Sciences, University of Zaragoza, 50009 Zaragoza, Spain; mcasausm@gmail.com (M.C.); lmoreno@unizar.es (L.A.M.); 2Instituto Agroalimentario de Aragón, 50013 Zaragoza, Spain; 3Instituto de Investigación Sanitaria Aragón, 50009 Zaragoza, Spain; mjarreta@unizar.es; 4Centro de Investigación Biomédica en Red de Fisiopatología de la Obesidad y Nutrición, 50009 Zaragoza, Spain; 5Department of Socio-sanitary Sciences, University of Murcia, 30100 Murcia, Spain; aurluna@um.es (A.L.M.); aurelio.luna@um.es (A.L.R.-C.); 6Faculty of Medicine, University of Zaragoza, 50009 Zaragoza, Spain

**Keywords:** SARS CoV-2, vaccination, overweight, obesity, vaccine adverse events

## Abstract

COVID-19 vaccines have shown high efficacy, with most side effects being mild–moderate and more frequently reported by females and people at younger ages. Since no studies have assessed the impact that weight status could have on the reported adverse reactions, we aim to study the association between weight status and reported side effects. We included data on 2136 adults from an online survey conducted from 6 May to 9 June 2021. The questionnaire was filled in by participants over Google forms. Generalized Linear Mixed Models were used. A higher risk of presenting fever ≥38°, vomiting, diarrhea and chills was found in those with a non-overweight status compared to those overweight after adjusting for age, sex, education, medication to prevent/relieve post-vaccination effects and vaccine administered. When adjusting, most of the significant effects, in the association between side effects of the COVID-19 vaccine and weight status, did not remain significant. In conclusion, a non-overweight status was associated with a higher risk of presenting fever ≥38°, vomiting, diarrhea and chills compared to those overweight. Nevertheless, most of the reported side effects to COVID-19 vaccine were not associated with a higher risk of presenting more adverse effects, and individual differences were determined by sex and age.

## 1. Introduction

Coronavirus disease 2019 (COVID-19) has caused over four million deaths worldwide [1]. In this pandemic context, immunization of the population through vaccination is a public health priority [2]. Different studies have shown the efficacy of the vaccines currently on the market, reporting levels above 90% for Pfizer, Moderna and Sputnik V or above 60% in the case of AstraZeneca/Vaxzevria or Janssen [3,4]. 

Male patients, elderly, obese and those with an excess visceral fat have shown to have a higher risk for the development of complications following COVID-19 infection [5]. Sex, age and co-morbidities may also determine differences in vaccine response [6]. In a recent study, it was suggested that side effects of COVID-19 vaccines are a product of a short burst of Type I interferon (IFN-I) generation together with the induction of an effective immune response [7]. Thus, side effects can vary considerably according to the recipient’s age and sex, with more severe effects in females than males and in younger people than in the elderly [8].

What researchers have not yet identified is whether there is any relationship between the initial inflammatory reaction and the long-term response that leads to protection. In fact, there is no study proving that those that have experienced more side effects from the COVID-19 vaccine are having higher levels of antibody titers and therefore a higher protection. However, younger people and female patients have reported stronger side effects with higher antibody titers than older age groups and males, respectively [8,9].

A latest work found that central obesity, smoking habit and hypertension were associated with lower antibody titers in response to COVID-19 mRNA vaccine [6]. Since obesity is a strong predictor of mortality among patients with COVID-19, it is important to take these data into account to develop future vaccine strategies for COVID-19 [10].

To the best of our knowledge, no studies have investigated the relationship between weight status (underweight, normal weight, overweight and obesity) with severity of side effects. Therefore, the present study aims to analyze the association between weight status and side effects after adjusting for important covariates such as age, sex, education, medication to prevent or relieve post-vaccination side effects and vaccine administered.

## 2. Materials and Methods

### 2.1. Study Design and Participants

Data were collected from an online anonymous survey conducted from 6 May to 9 June 2021. This cross-sectional study included a convenience sample of adult population Spain.

We used Google Forms, an online survey platform, to publish the questionnaire, and the link generated was then shared via social networks such as Facebook, Twitter and WhatsApp. The interviewees visited the URL on their electronic devices to answer the questionnaire. The inclusion criteria were individuals who (1) were 18 years old or older, (2) voluntarily agreed to participate in the online survey and (3) were able to read and complete the self-administered questionnaire independently.

After excluding those participants who did not complete more than 50% of the required questions (*n* = 45), a total of 2136 participants were finally included in the present study. During data collection in Spain, around 40% of the participants had not received any vaccination; only 1189 had received at least the first dose, and 690 had received the necessary vaccines (two doses in case of Pfizer, Moderna or AstraZeneca/Vaxzevria or just one dose in case of Janssen or those who have had COVID-19 infection). The sampling technique in this dataset is convenience sampling.

This survey data was approved by the Ethics Committee of Aragon (CEICA), Spain (No. C.I. 422 PI21/195 and the Ethics Committee of the University of Murcia (ID: 3449/2021). The survey data was conducted according to the Declaration of Helsinki.

### 2.2. Measures

#### Sociodemographic and Vaccine-Related Information

Potential post-vaccination side effects were identified and covered in this survey after conducting a literature search on different databases including Pubmed, Science Direct and Google Scholar.

They included: fever (divided into less than 38° or 38° or more); headache; muscle pain (myalgia); chills; diarrhea; arm soreness, redness or swelling; nausea; vomiting; red, itchy, swollen or painful rash; loss of appetite; sweating; chills; enlarged lymph nodes; tiredness, sleepiness or dizziness; altered menstrual cycle and others (respondents needed to write down additional side effects experienced) [11,12,13].

The study questionnaire was divided into three main sections. The first section was designed to collect general information about the participants such as gender, age group, migrant status, socioeconomic status (SES) information that included study level and profession, self-reported weight and height and possible infection status with SARSCoV-2 with experienced symptoms. The second section focused on measuring the attitudes and intention to vaccinate against COVID-19. The third section focused on the COVID-19 vaccine-related data such as type and date of COVID-19 vaccine, possible side effects experienced after the first and the second jab, timing and duration of the side effects, medication to prevent or relieve post-vaccination side effects and information received before getting the vaccine about possible side effects and co-morbidities.

For descriptive purposes, body mass index (BMI, kg/m^2^) was calculated based on self-reported body weight and body height. BMI was categorized as: underweight (BMI < 18.50 kg/m^2^), normal weight (BMI 18.50 to 24.99 kg/m^2^), overweight (BMI ≥ 25.00 to 29.90 kg/m^2^) and obesity (BMI ≥ 30.00 kg/m^2^).

As stated above, we also collected sociodemographic information such as gender (male vs. female), age group (18–25, 26–35, 35–45, 45–55, 56–65 and 65+), migrant status (born in Spain: yes vs. no), educational level (undergraduate, health sciences-related graduate or post-graduate, i.e., medicine, nursing, pharmacy, non-health sciences-related graduate or post-graduate) and occupation status (healthcare professional, non-healthcare professional, retired and unemployed or student).

Moreover, we included some questions about receiving flu vaccine in the previous year; possibility of vaccination against COVID-19 (“no, for medical reasons” (i.e., the hematologist recommended not to have it); “no, because I was pregnant”; “no, because they did not offer me the vaccine yet”; “no, because I just had COVID-19”; “no, because I refused to”; “yes, but I only got one dose (and I need two doses)”; “yes, and I got all doses” (i.e., one for Janssen or two for Pfizer, Moderna or AstraZenca/Vaxzevria)); and possibility of being vaccinated after knowing previously possible side effects (“yes, of course”; “yes, but I would think about it more”; ”I will have doubts”; “no, side effects that I had do not make up for it”).

### 2.3. Statistical Analysis

For descriptive analyses, numbers and percentage were used to detail sample characteristics. Percentages and chi-square tests were used to evaluate the associations between weight status and possible side effects during the first and the second dose.

Subsequently, we conducted Generalized Linear Mixed Models to study the associations between weight status (independent variable) and possible side effects (dependent variables) during the first and the second doses. When dividing into four categories (underweight, normal weight, overweight and obesity used as reference), models did not run properly, since a low number of cases were included in some categories. Therefore, analyses were also conducted combining non-overweight group (BMI < 25) vs. overweight group (BMI ≥ 25 used as reference). For the above regression, odds ratio (OR) and the respective 95% confidence intervals (CI) were estimated. All analyses were performed using SPSS version 26.0 (IBM Corporation, New York, NY, USA). The alpha level was set at 0.05, and *p* < 0.05 was considered statistically significant.

Raw models were unadjusted models, and adjusted models included gender, sex, educational status, medication to prevent or relieve post-vaccination side effects and vaccine administered. Since occupational status and education were highly correlated, educational status was not included as a confounding factor.

## 3. Results

Table 1 presents the characteristics of the sample of this study: 67.7% of the sample were female, 33.2% of the sample were graduate or post-graduate in a health science field, and 31.1% had a healthcare occupation. Table 2 and Appendix A display COVID-19 vaccine information and side effects. In June 2021, 55.6% reported to have been vaccinated against COVID-19 (27.6% with necessary doses); 44.3% were not vaccinated yet due to medical reasons (0.6%), to pregnancy (0.8%), because they refused to (1.5%), because they had just COVID-19 (2.8%) or because the vaccine had not been offered yet (38.6%). We found 1.9% of the sample were underweight, 58.0% normal weight, 28.1% overweight and 8.7% obese. Administrated vaccines were Pfizer (47.8%), AstraZeneca/Vaxzevria (38.6%), Moderna (10.4%) and Janssen (1.5%). Common reported side effects in the first and the second doses were arm soreness, redness or swelling (38.9% vs. 32.9%); headache (27.1% vs. 22.9%); chills (19.3% vs. 13.2%); myalgia (11.7% vs. 24.6%); fever < 38° (11.1% vs. 13.0%); fever ≥ 38° (11.7% vs. 6.8%); tiredness, sleepiness or dizziness (10.9% vs. 10.5%); loss of appetite (11.7% vs. 3.2%); sweating (7.8% vs. 3.2%); nausea (5.7% vs. 6.1%); enlarged lymph nodes (2.4% vs. 3.6%); diarrhea (1.8% vs. 1.9%); vomiting (1.2% vs. 1.9%); red, itchy, swollen or painful rash (1.3% vs. 1.2%); altered menstrual cycle (0.7% vs. 0.5%) and others (Guillain–Barre syndrome, herpes zoster emergence, allergy, low mood, dysgeusia, syncope, Pityriasis rosea). Regarding the severity of the symptoms, most of our sample reported that they had been stronger in the second dose than in the first dose (40.3%), while 30% reported to not have had any symptoms at any dose. In order to prevent or relieve post-vaccination side effects, 62.7% affirmed to have taken medication (most frequently paracetamol and ibuprofen). Following the question of being vaccinated after knowing side effects experienced, 88.9% had no doubts at all, 3.7% expressed doubts and 0.9% believed that side effects experienced did not make up for it.

Females and younger adults presented a higher percentage of side effects in both doses compared to males and elderly, respectively (data not shown). Regarding age, during the second dose in those aged 18–25 years old compared to those over 65, arm soreness, redness or swelling was presented by 77.8% vs. 16.5% of participants, respectively; headache (27.1% vs. 22.9%); chills (37.0% vs. 5.5%); myalgia (44.4% vs. 7.9%); fever < 38° (18.5% vs. 7.1%); fever ≥ 38° (14.8% vs. 1.6%); tiredness, sleepiness or dizziness (25.9% vs. 7.1%); loss of appetite (11.1% vs. 1.6%); sweating (14.8% vs. 0.0%); nausea (5.7% vs. 6.1%) or diarrhea (3.7% vs. 0.8%).

Table 3 and Table 4 show crosstabs of side effects after receiving the first dose and the second dose, respectively, of the vaccine against COVID-19 per weight status (in percentage). Most side effects were reported at a higher percentage in those who were underweight or normal weight compared to overweight or obese. In the first dose, the following symptoms were reported by respondents with underweight and obesity, respectively: fever < 38° (20.7% vs. 6.4%); fever ≥ 38° (31.0% vs. 6.4%); myalgia (48.3% vs. 12.8%); arm soreness, redness or swelling (51.7% vs. 28.4%); headache (44.8% vs. 18.3%); loss of appetite (10.3% vs. 0.0%); sweating (27.6% vs. 5.5%) and chills (9.8% vs. 12.5%). In the second dose, the following symptoms were reported by respondents with underweight and obesity, respectively: fever < 38° (25% vs. 10.9%); fever ≥ 38° (8.3% vs. 1.8%); myalgia (58.3% vs. 20.0%); arm soreness, redness or swelling (41.7% vs. 32.7%); headache (58.3% vs. 25.5%); loss of appetite (16.7% vs. 5.5%); sweating (25.0% vs. 3.6%) and chills (33.3% vs. 10.9%).

Table 5 presents the associations between side effects experienced in the first and the second doses of COVID-19 vaccine and weight status (underweight/normal weight compared to overweight/obese used as reference).

Results from Multinomial Logistic Regression of unadjusted models showed that fever ≥ 38° was more than twice as prevalent in respondents with non-overweight (underweight and normal weight) compared to those with overweight (including those with obesity), (OR = 2.67, 95% CI: 1.74–4.10). Similarly, in the second dose, fever ≥ 38° was also more prevalent in respondents with non-overweight (OR = 2.69, 95% CI: 1.25–5.77). After adjusting for sex, age, education, medication to prevent or relieve post-vaccination side effects and vaccine administered, results remained significant (OR = 2.54, 95% CI: 1.04–6.22). Non-overweight status was also associated with a higher risk of myalgia (OR = 2.49, 95% CI: 1.83–3.39) in the first and the second doses, significant (OR = 1.70, 95% CI: 1.14–2.55) compared to those with overweight. Nevertheless, this association was no longer significant in full-model adjustment. Similarly, in the first dose, non-overweight status was also associated with a higher risk of arm soreness, redness or swelling (OR= 1.59, 95% CI: 1.24–2.04); nausea (OR = 2.54, 95% CI: 1.39–4.65); loss of appetite (OR = 3.02, 95% CI: 1.45–6.29); sweating (OR = 2.69, 95% CI: 1.60–4.52) and tiredness, sleepiness or dizziness (OR = 1.60, 95% CI: 1.08–2.36). However, this association was not significant in fully adjusted models. In the second dose, non-overweight status was associated with vomiting (OR = 8.64, 95% CI: 1.97–37.92), diarrhea (OR = 2.85, 95% CI: 1.02–7.91) and chills (OR = 2.36, 95% CI: 1.19–4.67) compared to overweight status in fully adjusted models. No other significant results were found.

Appendix A show the side effects in the first and the second doses per COVID-19 vaccine (in percentage), respectively. In the first dose, AstraZeneca/Vaxzevria had the highest percentage of side effects compared to the rest of vaccines (Janssen, Moderna or Pfizer). In contrast, people vaccinated with Pfizer presented the lowest percentage of side effects. In the second dose, Moderna presented a higher percentage of side effects compared to Pfizer and AstraZeneca/Vaxzevria.

Appendix A present the associations between side effects experienced in the first and second doses of COVID-19 vaccine and weight status (underweight, normal weight, overweight and reference: obese). The results from the Multinomial Logistic Regression Models showed that side effects (myalgia; arm soreness, redness or swelling; headache and chills) were in a higher percentage when decreasing weight status in raw models. That is, side effects were more prevalent in those who were underweight and normal weight compared to those who were obese. However, no results remained significant when using adjusted models. Similar results were found when conducting sensitivity analyses (Appendix A) to explore the association between weight status and side effects for the first dose of the COVID-19 vaccine in the population receiving Pfizer (*n* = 568) and, in the population, receiving AstraZeneca/Vaxzevria (*n* = 459).

## 4. Discussion

The present study aimed to study the association between possible side effects after receiving a COVID-19 vaccine and weight status. We found a higher risk of presenting some side effects against COVID-19 vaccine (fever ≥ 38°, vomiting, diarrhea and chills) in those with a non-overweight status (underweight and normal weight) compared to those overweight (including obesity) after adjusting for important covariates such as age, sex, education, medication to prevent or relieve post-vaccination and vaccine administered. Most of the side effects experienced were significantly higher in those who were non-overweight compared to those overweight. Nonetheless, sex and age were the most important variables to predict possible side effects experienced against COVID-19 vaccine. In fact, when adjusting for sex and age, most of the significant effects, in the association between side effects of the COVID-19 vaccine and weight status, did not remain significant. These results are in agreement with previous works that reported lower adverse effects in males and those aged over 55 years [14,15,16].

In our sample, most of the people who were overweight or obese were also those who were older (over 55 years), which could partly explain why a BMI over 25 was related to lower side effect experienced in our study.

Other factors such as type of COVID-19 vaccine and dose received might also play an important role [14,15,16]. Those who received a chimpanzee adenovirus-vectored vaccine such as AstraZeneca/Vaxzevria reported more commonly adverse reactions than those who received a mRNA-based vaccine such as Pfizer or Moderna (and among these two vaccines, those who received Moderna experienced more adverse reactions than those who received Pfizer). We have to bear in mind that, in the case of AstraZeneca/Vaxzevria, during the first months of use (February and March and mid-April 2021), health authorities in Spain recommended inoculation in the population under 55 years of age. However, after some studies suggested a possible link of blood clots with low blood platelets due to the AstraZeneca/Vaxzevria vaccine [17], health authorities in Spain prohibited it in people under 60 years old [18]. Those who had received AstraZeneca/Vaxzevria in the first dose could choose between being vaccinated with AstraZeneca/Vaxzevria or Pfizer in the second dose after the recommended period between doses (at least three months). This fact made side effects stronger after the first dose compared to those after the second dose, partly because only younger adults were eligible for receiving the former. Currently, only people who are 60 or older can be vaccinated with AstraZeneca/Vaxzevria. Hence, a lower number of people received vaccinated with AstraZeneca/Vaxzevria vaccine in both doses. Furthermore, regarding the age-based COVID-19 vaccine strategy, Spain started the vaccination with those at a higher risk of complications (>80 years old). Therefore, in the present study, mean age of people with both doses was higher compared to those who had only one dose. As younger people generally have a more a robust immune system, the immune response to the vaccine and the reported side effects are higher.

To the best of our knowledge, no previous investigations have studied a relationship between side effects experienced against COVID-19 vaccine and weight status. Although a recent study that aimed to investigate the safety and effectiveness of these vaccines in a UK community setting and used obesity to adjust all models found some differences between strata of BMI, there was no clear trend across vaccines and doses [14]. However, previous studies investigating side effects with other vaccines (particularly with the Anthrax vaccine) suggested that obese women may experience elevated arm soreness, and a depressed pre-vaccination serum progesterone level may increase a woman’s odds of experiencing injection site arm swelling [19]. Similarly, in another study, redness and swelling were more common for obese participants compared to participants who were not overweight [20]. A higher reaction in the infection-site could be caused by inadvertent sub-cutaneous administration rather than intra-muscularly [21]. Contrary to these studies, Petousis-Harris et al. showed that those with a lower BMI reported more pain when meningococcal B vaccine was administrated [22].

Although obesity is associated with a low-level chronic inflammation [23], a higher reactogenicity in the overweight population seemed to be linked to the vaccine administration technique and not to BMI, itself [24]. In a study of overweight/obese participants and in normal weight controls, the frequency of injection-site and systemic reactions after trivalent influenza vaccine was statistically similar between groups, which is in agreement with our results. Thus, for the adverse reaction of arm soreness, redness or swelling, individual injection technique (that depends on the vaccinator) could be the reason behind observed differences in some studies [25].

No other side effects such as fever; myalgia; diarrhea; tiredness, sleepiness or dizziness; nausea or vomiting have been studied in other vaccines regarding BMI status. Investigations have not identified any relationship between the initial inflammatory reaction and the long-term response that leads to protection. Hence, individuals with stronger side effects from the vaccine are not better protected from COVID-19 [26]. Nevertheless, younger people and females experience higher side effects due to a more a robust immune system and produce higher antibody titers after being vaccinated [27]. In contrast, central obesity as well as smoking habit and hypertension were associated with lower antibody titers in response to COVID-19 mRNA vaccine [6]. Lower adverse effects experienced after receiving COVID-19 vaccine may explain why those obese had lower antibodies titers and lower percentage of some side effects against COVID-19 vaccine (fever ≥ 38°, vomiting, diarrhea and chills) compared with non-overweight individuals after adjusting for important covariates such as age, sex, education, medication and vaccine administered to prevent or relieve post-vaccination.

To our knowledge, this is the first study conducted in Spain to investigate the association between possible side effects after receiving a COVID-19 vaccine and weight status with a sample size of 2136 individuals (of which 31.1% were healthcare professionals) after the approval of the vaccine. In addition, although we used a convenience sampling, we have a good representation of different age groups, and women represented 67.7% of the respondents which is in agreement with the percentages of healthcare professionals in Spain and worldwide [28].

Nevertheless, our investigation has several limitations. This study is not random and therefore is not representative of the Spanish population. Moreover, there are some groups that could be underestimated, in part due to the collection method used (i.e., males represented 32.3% of the sample, and migrants were 6.4%). Hence, the extrapolation of these results can be difficult. In fact, although online questionnaires are simple tools that can offer advantages such as the access to different population and prompt answers, some questions that can arise when auto-filling the questionnaire and could be responded to in a face-to-face interview are difficult to address in online surveys. Besides, we used self-reported data for weight and height, the reliability and validity of which have been found conflicting by some studies [28]. However, a recent study revealed good agreement between self-reported and direct anthropometric measurements [29]. Additionally, it should be borne in mind that side effects experienced depend mainly on age and sex. Most of the fully vaccinated people included in the present sample were above 56 years old (46.1%), and thus, side effects could be underestimated. Finally, it should be noted that, although the total sample amounted to 2136 subjects, the effective sample of participants who received both doses was 590, which resulted in the impossibility of performing certain sub-group analyses. Consequently, results should be interpreted and considered on the bases of all the above.

## 5. Practical Implications

Previous investigations have demonstrated that people who are obese are more likely to have severe COVID-19 outcomes [5]. Nonetheless, the present paper has shown that overweight or obese individuals are not at a higher risk of presenting adverse events compared to those underweight or normal weight. A recent study indicated the antibody levels are lower in a person who is overweight or obese as compared to a person of normal weight [6]. This speculation about variable effectiveness of COVID-19 vaccines in obesity likely increases vaccine hesitancy among individuals with obesity. Considering that adverse effects were found to be mild or moderate and that there is likelihood that the levels of antibody (and subsequent immunity) are still high enough to protect against COVID-19, our results suggest the need for being vaccinated among people with overweight or obesity.

## 6. Conclusions

The success of COVID-19 vaccination depends, to a large extent, on the confidence that the population has in vaccines. To date, the current vaccines against COVID-19 have shown high efficacy and are expected to perform well in terms of side effects. However, the scarcity of post-vaccination studies makes it advisable to promote research that contributes to a deeper understanding of the results of vaccination in practice and counteracts the risks of a priori loss of confidence in certain vaccines on the part of the population. Mostly all reported side effects from the COVID-19 were mild or moderate. Sex and age were the most important variables to predict possible side effects experienced against COVID-19 vaccine. We also found a higher risk of presenting some side effects against COVID-19 vaccine (fever ≥ 38°, vomiting, diarrhea and chills) in those with a non-overweight status (underweight and normal weight) compared to those overweight (including obesity) after adjusting for important covariates such as age, sex, education, medication to prevent or relieve post-vaccination and vaccine administered. Nonetheless, when adjusting for sex and age, most of the significant effects, in the association between side effects of the COVID-19 vaccine and weight status, did not remain significant.

## Figures and Tables

**Table 1 vaccines-09-01321-t001:** Sample characteristics (*n* = 2136).

*n* = 2136	N	%
**Age (in years)**		
18–25	307	14.4
26–35	223	10.4
36–45	526	24.6
46–55	480	22.5
56–65	369	17.3
>65	231	10.8
**Gender**		
Male	690	32.3
Female	1446	67.7
**Education**		
Undergraduate	599	28.0
Graduate or Post-graduate (non-health sciences-related)	827	38.7
Graduate or Post-graduate (health sciences-related)	710	33.2
**Weight status**		
Underweight	41	1.9
Normal weight	1244	58.0
Overweight	604	28.1
Obesity	187	8.7
Missing	60	2.8

**Table 2 vaccines-09-01321-t002:** COVID-19 vaccine side effects (*n* = 1189).

** *n* ** ** = 1189 vaccinated with at least one dose**	** *n* **	**%**
**Administrated vaccine (first dose)**		
AstraZeneca/Vaxzevria	459	38.6
Janssen	18	1.5
Moderna	124	10.4
Pfizer	568	47.8
Missing	20	1.7
**Side effects first dose**		
**Fever < 38°**		
Yes	132	11.1
No	1042	87.6
Missing	15	1.3
**Fever ≥ 38°**		
Yes	139	11.7
No	1035	87.0
Missing	15	1.3
**Myalgia (muscle pain)**		
Yes	139	11.7
No	1035	87.0
Missing	15	1.3
**Arm soreness, redness or swelling**		
Yes	462	38.9
No	712	59.9
Missing	15	1.3
**Nausea**	497	23.3
Yes	68	5.7
No	1106	93.0
Missing	15	1.3
**Vomiting**		
Yes	14	1.2
No	1160	97.6
Missing	15	1.3
**Red, itchy, swollen or painful rash**		
Yes	15	1.3
No	1154	97.1
Missing	20	1.7
**Headache**		
Yes	322	27.1
No	852	71.7
Missing	15	1.3
**Diarrhea**		
Yes	21	1.8
No	1153	97.0
Missing	15	1.3
**Loss of appetite**		
Yes	139	11.7
No	1035	87.0
Missing	15	1.3
**Sweating**		
Yes	93	7.8
No	1081	90.9
Missing	15	1.3
**Chills**		
Yes	230	19.3
No	944	79.4
Missing	15	1.3
**Enlarged lymph nodes**		
Yes	28	2.4
No	1146	96.4
Missing	15	1.3
**Altered menstrual cycle**		
Yes	13	1.1
No	1161	97.6
Missing	15	1.3
**Tiredness, sleepiness or dizziness**		
Yes	130	10.9
No	1154	97.1
Missing	15	1.3
**Medication to prevent or relieve post-vaccination side effects**		
Yes	746	62.7
No	443	37.3

**Table 3 vaccines-09-01321-t003:** Crosstabs of side effects after receiving the first dose of the vaccine against COVID-19 per weight status (in percentage).

%	Fever < 38°	Fever ≥ 38°	Myalgia	Arm Soreness, Redness, Swelling	Nausea	Vomiting	Red, Itchy, Swollen or Painful Rash	Headache	Diarrhea	Loss of Appetite	Sweating	Chills	Enlarged Lymph nodes	Altered Menstrual Cycle	Tiredness, Sleepiness, Dizziness
Underweight	20.7	31.0	48.3	51.7	20.7	3.4	3.4	44.8	0.0	10.3	27.6	9.8	0.0	0.0	24.1
Normal weight	12.6	14.7	29.3	43.6	6.9	1.5	2.0	32.5	2.0	5.6	9.8	41.4	2.9	1.4	12.9
Overweight	10.1	6.4	15.4	35.4	3.2	0.9	1.2	19.4	1.7	2.6	3.8	23.8	2.3	1.2	8.4
Obese	6.4	6.4	12.8	28.4	2.8	0.0	1.8	18.3	1.8	0.0	5.5	12.5	0.9	0.0	0.9
Significant differences	*	*	*	*	*		*	*		*	*	*			*

* Significant results of Pearson Chi-Square.

**Table 4 vaccines-09-01321-t004:** Cross tabs of side effects after receiving the second dose of the vaccine against COVID-19 (N = 590) per weight status (in percentage).

%	Fever < 38°	Fever ≥ 38°	Myalgia	Arm Soreness, Redness, Swelling	Nausea	Vomiting	Red, Itchy, Swollen or Painful Rash	Headache	Diarrhea	Loss of Appetite	Sweating	Chills	Enlarged Lymph Nodes	Altered Menstrual Cycle	Tiredness, Sleepiness, Dizziness
Underweight	25.0	8.3	58.3	41.7	8.3	8.3	0.0	58.3	0.0	16.7	25.0	33.3	0.0	0.0	25.0
Normal weight	14.8	9.6	28.3	35.0	2.9	2.9	1.6	23.2	2.6	2.9	2.9	17.0	4.2	0.6	9.0
Overweight	13.2	4.4	19.2	30.2	3.3	0.5	1.1	20.3	0.5	1.6	1.6	6.6	2.2	0.5	6.0
Obese	10.9	1.8	20.0	32.7	9.1	0.0	0.0	25.5	1.8	5.5	3.6	10.9	5.5	0.0	10.9
Significant differences	*	*	*	*				*		*	*	*			*

* Significant results of Pearson Chi-Square.

**Table 5 vaccines-09-01321-t005:** Associations between weight status (reference: overweight/obese) and side effects experienced in the first and the second doses of COVID-19 vaccine. Results from Multinomial Logistic Regression Models.

	First dose: Underweight/Normal Weight (M0 Unadjusted Model)		First Dose: Underweight/Normal Weight (M1 Adjusted for Age and Sex)		First Dose: Underweight/Normal Weight (M2 Adjusted for Age, Sex, Education Vaccine Administered and Medication)		Second Dose: Underweight/Normal Weight (M0 Unadjusted Model)		Second Dose: Underweight/Normal Weight (M1 Adjusted for Age and Sex)		Second Dose: Underweight/Normal Weight (M2 Adjusted for Age, Sex, Education, Vaccine Administered and Medication)	
	OR	95% CI	OR	95% CI	OR	95% CI	OR	95% CI	OR	95% CI	OR	95% CI
Fever < 38°	1.46	0.99–2.15	0.98	0.61–1.59	0.90	0.58–1.38	1.46	0.99–2.15	1.24	0.76–2.01	1.21	0.68–2.12
Fever ≥ 38°	**2.67**	**1.74**–**4.10**	0.94	0.50–1.78	0.86	0.49–1.52	**2.69**	**1.25**–**5.77**	2.23	0.78–6.39	**2.54**	**1.04**–**6.22**
Myalgia	**2.49**	**1.83**–**3.39**	1.28	0.81–2.01	1.22	0.80–1.86	**1.70**	**1.14**–**2.55**	1.26	0.83–1.92	1.38	0.90–2.11
Arm soreness, redness, swelling	**1.59**	**1.24**–**2.04**	0.98	0.73–1.33	0.98	0.71–1.35	1.22	0.86–1.75	0.89	0.65–1.22	0.92	0.60–1.41
Nausea	**2.54**	**1.39**–**4.65**	0.72	0.16–3.29	1.51	0.72–3.15	1.20	0.59–2.45	1.01	0.59–1.73	1.03	0.56–1.91
Vomiting	2.42	0.67–8.72	0.68	0.06–7.63	1.69	0.41–6.89	7.54	0.95–59.57	6.40	0.94–43.64	**8.64**	**1.97**–**37.92**
Red, itchy, swollen or painful rash	1.53	0.58–4.02	1.11	0.16–7.42	1.05	0.38–2.92	1.85	0.35–9.64	1.50	0.87–2.58	1.56	0.73–3.38
Headache	**2.08**	**1.57**–**2.76**	1.06	0.74–1.51	0.99	0.68–1.44	1.18	0.79–1.76	0.84	0.51–1.40	0.82	0.51–1.30
Diarrhea	1.06	0.43–2.59	0.78	0.23–2.62	0.74	0.36–1.51	2.98	0.62–14.23	2.96	0.54–16.22	**2.85**	**1.02**–**7.91**
Loss of appetite	**3.02**	**1.45**–**6.29**	1.56	0.53–4.55	1.45	0.55–3.83	1.36	0.49–3.73	0.74	0.43–1.25	0.79	0.24–2.55
Sweating	**2.69**	**1.60**–**4.52**	1.11	0.22–5.68	0.87	0.52–1.45	1.79	0.62–5.16	1.22	0.60–2.46	1.13	0.74–1.72
Chills	**2.31**	**1.66**–**3.21**	1.08	0.67–1.75	1.01	0.67–1.52	**2.61**	**1.49**–**4.57**	1.97	1.14–3.40	**2.36**	**1.19**–**4.67**
Enlarged lymph nodes	1.39	0.62–3.10	0.69	0.28–1.73	0.67	0.39–1.15	1.38	0.54–3.51	1.00	0.45–2.23	0.84	0.39–1.80
Altered menstrual cycle	1.47	0.45–4.83	0.67	0.20–2.25	0.64	0.34–1.20	0.67	0.20–2.25	0.67	0.20–2.25	0.99	0.68–1.44
Tiredness, sleepiness, dizziness	**1.60**	**1.08**–**2.36**	1.04	0.27–3.92	0.79	0.58–1.09	1.37	0.74–2.55	1.04	0.27-3.92	1.66	0.95–2.89

Statistically significant results shown in bold font; M0: unadjusted models; M1: model adjusted for age and sex; M2: model adjusted for age, sex, education status, medication to prevent or relieve post-vaccination side effects and vaccine administered (full-adjusted models).

## Data Availability

The raw data supporting the conclusions of this article will be made available by the authors upon request, without undue reservation.

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
