# Peer review of "Association between COVID-19 Vaccine Side Effects and Body Mass Index in Spain"

_vaccines, 2021, doi:10.3390/vaccines9111321_

Round 1

Reviewer 1 Report

The study “Associations between COVID-19 vaccine side effects and body mass index in Spain"  analyzed the association between weight status and side effects after adjusting for important covariates such as age, sex, education, and medication to prevent or relieve post-vaccination side effects. There are some issues needed to be noticed.

Major comments

  1. A total of 2,136 participants were finally included in the present study, while only 1,189 had received at least the first dose, and 590 had received two doses. Is the effective sample size sufficient for analysis?
  2. This study aimed to assess the association between weight status and side effects of the COVID-19 vaccine after adjusting for covariates such as age, sex, education, and medication to prevent or relieve post-vaccination side effects. However, this study included four types of vaccines, each with different side effects and different incidence of side effects. Therefore, it is not appropriated to explore the association between weight status and side effects in this study.
  3. Age and sex were significantly associated with BMI, while other studies showed that side effects were associated with age and sex. BMI may be a confounding factor.
  4. Table 1 needs to be rearranged since it is poorly readable.

Reviewer 2 Report

This cross-sectional online survey study was conducted in Spain to investigate the attitudes towards vaccines and intention to vaccinate against COVID-19. Several demographic and socioeconomic characteristics were taken into account with a sample size of 2,136 individuals (of which 270 31.1% were healthcare professionals). The study carries several limitations for example it is not random and therefore is not representative of the Spanish population. Moreover, there are some groups that could be underestimated, in part due to the collection method. Therefore, the extrapolation of these results is not easy. In addition, side effects experienced depend mainly on age and sex. Most of the fully vaccinated people were above 56 years old (46,1%) and therefore side effects could be underestimated. The authors have noted the limitations and based on the findings they have concluded that the success of COVID-19 vaccination depends, to a large extent, on the confidence that the population has in vaccines. majority of the respondents reported side effects from the COVID-19 were mild or moderate. Sex and age were the most important variables to predict possible side effects experienced against COVID-19 vaccine. A higher risk of some side effects against COVID-19 vaccine (fever ≥38º, vomiting, diarrhea and chills) was seen in those with a non-overweight status compared with those overweight (including obesity) after adjusting for important covariates such as age, sex, education and medication. Following the adjusting, for sex and age most of the significant effects, in the association between side effects of the COVID-19 vaccine and weight status, did not stay significant.

Please proofread the text for English. 

Please check with the text size after tables, it seems it became smaller in the main text as it should not be. 

Please add a recommendation as to what can be used for out of the findings of this study, for example, in terms of use by clinicians, researchers, and individuals in the society. 

Reviewer 3 Report

General comment: The authors presented an interesting and original work addressing the association between COVID-19 vaccine side effects and body mass index in Spain. For this, data from 2,136 adults was obtained through online surveys.

Title: In my opinion, the title should be improved to “Association between COVID-19 vaccine side effects and body mass index in Spain”.

Abstract: It is complete and structured adequately.

Keywords: The keywords should be different from those used in the title. 

Introduction: An adequate and complete contextualization of the theme was performed in this section. The aim was clearly stated in this section. 

Materials and methods: The materials and methods are adequately described. The authors indicated the methods of data collection, as well as the exclusion and inclusion criteria. The statistical analysis employed was also described.

Results: The results are supported by the tables.

The table 1 is too long, which makes it confusing. It should be divided. One table with population characterization (age, gender, nationality, education,…), one table with COVID data (vaccination against COVID, infection, possibility of being vaccinated,…) and one table with side effects.

Please check the percentage in the following sentence: “33.1% of the sample were graduate or postgraduate in a Health Science field…” In the table 1 is indicated 33.2% for graduate or postgraduate (Health Sciences related).

The authors said: “55,2% reported to have been vaccinated against COVID-19 (27.6% with necessary doses). But, in the percentage of vaccinated people indicated in table 1 is 55.6% (27.6 + 28.0)”.

“44.8% were not vaccinated yet due to medical reasons (0.6%), pregnancy (0.8%), had just COVID-19 (0.5%) or the vaccine had not been offered yet (38.6%).” Please check these percentages.

“55,2% reported to have been vaccinated against 141 COVID-19 (27.6% with necessary doses).” Please check the percentage 27.6. According to the table 1, 28.0% of the people were vaccinated and got all doses.

Please check the percentages in the followings sentence: “44.8% were not vaccinated yet due to medical 142 reasons (0.6%), pregnancy (0.8%), had just COVID-19 (0.5%) or the vaccine had not been 143 offered yet (38.6%).” They are not in accordance with the table 1.

Several mistakes on the percentages indicated in the text occurred throughout the document. Please check all of them.

Please check the following sentence for English spelling: “1.5% of the sample affirmed that they vaccine had been offered...”

Discussion: The results are adequately discussed and compared with those observed by other authors.

Recommendation: The manuscript should not be accepted for publication in the present form. It should be reconsidered after a major revision.

Round 2

Reviewer 1 Report

The author had addressed some of our previous major comments. However, two issues remained to be addressed.

  1. Please add types of vaccines as a covariate in the adjusted model in table 5.
  2. Please add two sensitivity analyses to explore the association between weight status and side effects for the first and second doses of COVID-19 vaccine: 1) in the population receiving Pfizer(568); 2) in the population receiving AstraZeneca/Vaxzevria(459). 

Author Response

Reviewer #1:

The author had addressed some of our previous major comments. However, two issues remained to be addressed.

Please add types of vaccines as a covariate in the adjusted model in table 5.

Answer: Thank you very much for your suggestion. We have conducted the pertinent analyses and we have added the new results in the manuscript.

Please add two sensitivity analyses to explore the association between weight status and side effects for the first and second doses of COVID-19 vaccine: 1) in the population receiving Pfizer (568); 2) in the population receiving AstraZeneca/Vaxzevria (459).

Answer: As suggested, we have included two sensitivity analyses to explore the association between weight status and side effects for the first dose of the COVID-19 vaccine in the population receiving Pfizer and, in the population, receiving AstraZeneca/Vaxzevria. For the second dose we could not include these analyses due to the small sample size that caused unstable models. These results have been added as supplementary material (Table S6).

Reviewer 3 Report

Once the comments were addressed and the manuscript was improved, it should be accepted for publication.

Author Response

Once the comments were addressed and the manuscript was improved, it should be accepted for publication.

Answer: Thank you for your time and previous comments.